# Effect of Acute Heat Stress on the mRNA Levels of Cytokines in Broiler Chickens Subjected to Embryonic Thermal Manipulation

**DOI:** 10.3390/ani9080499

**Published:** 2019-07-29

**Authors:** Khaled M. M. Saleh, Mohammad B. Al-Zghoul

**Affiliations:** 1Department of Applied Biological Sciences, Faculty of Science and Art, Jordan University of Science and Technology, P.O. Box 3030, Irbid 22110, Jordan; 2Department of Basic Medical Veterinary Sciences, Faculty of Veterinary Medicine, Jordan University of Science and Technology, P.O. Box 3030, Irbid 22110, Jordan

**Keywords:** broiler, thermal manipulation, interleukins, interferons, acute heat stress

## Abstract

**Simple Summary:**

Heat stress affects animal husbandry by impeding the health and production of livestock and leads to commercial losses during the hot summer season. The broiler chicken is one variety of poultry that is bred and reared for commercial meat production, but exposing this breed to high environmental temperatures causes immunosuppression. To improve heat stress tolerance in broilers, thermal manipulation (TM), a process which involves timed changes in incubation temperature during embryonic development, has been suggested as a way to enhance thermotolerance acquisition and immune response. Cytokines are small extracellular signaling peptides with critical roles in immunity by enabling cell communication during immunological development and an immune response. The objective of the current study was to investigate the effects of TM during broiler chicken embryonic development on splenic mRNA levels of the Interferon-α (IFN-α), Interferon- β (IFN-β), Interferon- γ (IFN-γ), Interleukin-4 (IL-4), Interleukin-8 (IL-8), Interleukin-15 (IL-15), Interleukin-16 (IL-16), Interleukin-17 (IL-17), and Interleukin-18 (IL-18) genes during acute heat stress (AHS). Our findings suggest that TM has a long-term effect on cytokine expression dynamics during AHS. Consequently, TM may improve heat tolerance acquisition by increasing the expression of signaling proteins important to tissue stability as well as to repair mechanisms that are employed during and/or after heat stress recovery.

**Abstract:**

Heat stress significantly impacts the immunity and cytokine expression of chickens. However, the effects of embryonic thermal manipulation (TM) on cytokine expression in broiler chickens (broilers) is unclear. The objective of the current study was to evaluate the effects of TM on the splenic mRNA expression dynamics of certain cytokines—namely, IFN-α, IFN-β, IFN-γ, IL-4, IL-8, IL-15, IL-16, IL-17, and IL-18—in broilers during subsequent exposure to acute heat stress (AHS). TM was performed by elevating the incubation temperature to 39 °C at 65% relative humidity (RH) for 18 h daily during embryonic days (ED) 10–18. On post-hatch day 28, AHS was carried out for 7 h at 40 °C. At 0 h and after 1, 3, 5, and 7 h of AHS, splenic tissues were collected from all study groups to evaluate mRNA expression by relative-quantitative real-time (RT)-PCR. Plasma was collected to measure IL-4, IL-8, and IFN-γ levels. At 0 h, TM significantly reduced the basal mRNA level of IFN-β and the plasma level of IFN-γ and IL-8. Moreover, AHS significantly decreased IFN-β in control chicks, decreased IL-4 in both TM and control chicks, and increased IFN-γ and IL-16 in TM chicks. IFN-α, IL-8, IL-15, IL-17, and IL-18 expression all significantly increased during AHS in both TM and control chicks, but expression dynamics were improved in TM chicks for all cytokines (except IL-17). AHS resulted in increased plasma IFN-γ levels in TM chicks only, and increased IL-8 levels at 3 and 5 h of AHS in TM chicks, but at 7 h in control chicks. Lastly, 3 h of AHS increased IL-4 plasma levels in control chicks. The results of this study may indicate that TM has a long-term effect on cytokine expression dynamics of broilers, especially during AHS. Therefore, TM may improve heat tolerance acquisition by increasing the expression of signaling proteins important to tissue stability and to repair mechanisms that are employed during and/or after heat stress recovery.

## 1. Introduction

Heat stress affects animal husbandry by impeding the health and production of livestock, leading to commercial losses during hot summer seasons [1]. In poultry, the complications associated with heat stress include reductions in feed intake, growth rates, hatchability rates, and meat production [2]. Exposing broiler chickens (broilers) to high environmental temperatures results in immunosuppression through the dual activation of the sympathetic nervous system alongside the hypothalamic–pituitary–adrenal axis, leading to high catecholamine and corticosterone serum levels [2,3,4]. Furthermore, heat stress was found to modulate the gene expression of a range of different cytokines in broiler chickens [5,6].

Cytokines are small extracellular signaling peptides which have been shown to play a critical role in immunity [7,8]. In fact, cytokines enable cell communication during immunological development and immune response, as every cell type is capable of secreting them [8,9]. The IL-1β, IL-18, IL-17, IL-6, IL-16, and IL-8 cytokines have been characterized in chickens as having pro-inflammatory functions, while IL-10 is an anti-inflammatory cytokine. With respect to avian immune function, IL-2, IL-15, and IL-21 were found to promote T-cell proliferation, IL-12 and IFN-γ enhanced cell-mediated immunity, IL-4 and IL-19 increased antibody-mediated immunity, and IFN-α, IFN-β, and IFN-λ displayed antiviral activity [10]. In addition to the above, cytokines play an important role in healing injured tissue, including insults arising as a result of heat stroke [11,12]. Specifically, heat stress has been found to modulate the splenic and intestinal gene expression of several different types of cytokines in chickens [5,13,14].

To alleviate heat stress in broilers, thermal manipulation has been suggested; this process involves timed changes in incubation temperature during embryonic development [15,16,17,18,19,20,21,22]. It has been postulated that TM improves broiler immune response during post-hatch life by reducing *Salmonella enteritidis* colonization as well as enhancing cell-mediated immune response during post-hatch exposure to chronic heat stress (CHS) [18,23,24]. In ducklings, TM during the middle stage of embryogenesis led to immunosuppression and reduced plasma IFN-γ levels [25], while TM during embryonic days (ED) 1 to 21 led to increased IL-6, IFN-γ, and IL-10 expression in Pekin duck embryos [26]. Furthermore, TM during ED 1 to 25 led to increased IL-10, IL-6, and IFN-γ expression in ducklings challenged with lipopolysaccharide (LPS), suggesting that TM may be an aid in the response to subsequent post-hatch inflammations [27].

Although cytokines are important for normal immune status in vertebrates, it is still unclear how TM affects their levels during post-hatch life in broiler chickens, especially during heat stress periods. Therefore, the objective of this study was to investigate the effects of TM during broiler embryonic development on splenic mRNA levels of the IFN-α, IFN-β, IFN-γ, IL-4, IL-8, IL-15, IL-16, IL-17, and IL-18 genes during acute heat stress (AHS).

## 2. Materials and Methods

Jordan University of Science and Technology’s Animal Care and Use Committee (JUST-ACUC) approved all experimental procedures and management conditions used in this study.

### 2.1. Egg Procurement and Incubation Conditions

Six hundred (600) fertile eggs belonging to the Hubbard broiler strain were obtained from a certified breeder based in Irbid, Jordan. The eggs were inspected for any breakage or abnormality, and eggs were excluded if they were broken, deformed, round, wrinkled, or outside the normal size range (55–70 g). Approved eggs were randomly divided and incubated in two semi-commercial Type-I HS-SF incubators (Masalla, Barcelona, Spain). The eggs were subdivided into two treatment groups: The control group (*n* = 293) and the thermal manipulation (TM) group (*n* = 295). Eggs in the control group were maintained under standard conditions (37.8 °C and 56% relative humidity (RH)) throughout the embryonic period, while the TM group was subject to TM at 39 °C and 65% RH for 18 h during ED 10–18. The RH was increased to 65% to avoid an increase in water evaporation from the eggs during TM. On ED 7, the eggs were examined by candling: Eggs with dead embryos and infertile eggs were removed.

### 2.2. Rearing and Exposure to Acute Heat Stress (AHS)

On hatching-day, the one-day-old feather-dried chicks were transferred to the Jordan University of Science and Technology’s Animal House for the performance of field experiments. Chicks were divided into groups of ten and randomly distributed into their cage pens. During the first week post-hatch, the room temperature was maintained at 33 ± 1 °C and 50–60% RH, which was gradually decreased to 24 °C and 50–60% RH by the end of the third week post-hatch. Starting from post-hatch day 24 to 28, the room temperature was maintained at 21 °C and 50–60% RH. Water and standard feed rations (Table 1) were provided to the chicks ad libitum throughout the experimental period. The chicks were vaccinated against Newcastle disease (after 8 and 20 days of age) and infectious bursal disease (on post-hatch day 15).

On day 28 post-hatch, 40 chicks each were randomly selected from the control and TM groups and subjected to acute heat stress (AHS). AHS was induced by raising the room temperature to 40 °C and 50–60% RH for a duration of seven hours. At zero hours of AHS and after 1, 3, 5 and 7 h of AHS, blood samples (*n* = 6) were collected from the brachial wing veins from each treatment group into EDTA tubes. Afterwards, these birds were humanely euthanized for splenic sample collection.

### 2.3. RNA Extraction and cDNA Synthesis

Total splenic RNA was isolated using Direct-Zol™ RNA MiniPrep (Zymo Research, Irvine, CA, USA) combined with TRI Reagent^®^ (Zymo Research, Irvine, CA, USA). RNA concentrations and purities were determined by the Biotek PowerWave XS2 Spectrophotometer (BioTek Instruments, Inc., Winooski, VT, USA), and cDNA synthesis was carried out by inputting 2 μg of total RNA from each sample into the Superscript III cDNA Synthesis Kit (Invitrogen, Carlsbad, CA, USA).

### 2.4. Real-Time RT-qPCR Analysis

IFN-α, IFN-β, IFN-γ, IL-4, IL-8, IL-15, IL-16, IL-17, and IL-18 mRNA gene expressions were measured using relative-quantitative real-time PCR (RT-qPCR). The QuantiFast SYBR^®^ Green PCR Kit (Qiagen, Hilden, Germany) was used in the Rotor-Gene Q MDx 5plex Platform (Qiagen, Hilden, Germany), according to a previously published protocol (Al-Zghoul et al., 2019). Triplicates from each cDNA library were analyzed. Table 2 shows the sequences of the primers that were used in the real-time RT-PCR analysis.

### 2.5. Plasma Analysis

All blood samples were centrifuged at 5000 g for 10 min. Plasma samples were collected and stored at −20 °C until further analyses. Plasma levels IL-4, IL-8, and IFN-γ were determined using a commercially available ELISA kit, according to the manufacturer’s instructions (Cusabio Biotech Co., Ltd., Wuhan, China; Cat. No.: CSB-E06756Ch for IL-4, CSB-E14191C for IL-8, and CSB-E08550Ch for IFN-γ).

### 2.6. Statistical Analysis

All statistical analyses were performed with the IBM SPSS Statistics 25 software (IBM, Chicago, IL, USA). IL-4, IL-8, and IFN-γ plasma levels and IL-4, IL-8, IL-15, IL-16, IL-17, IL-18, IFN-α, IFN-β, and IFN-γ mRNA levels were expressed as means ± SD. Two-way ANOVA, followed by the Bonferroni test, was used to compare different parameters between the TM and control groups and to compare within treatment groups (between different time intervals of AHS: 0, 1, 3, 5, and 7 h). Parametric differences were considered statistically significant at *p* < 0.05.

## 3. Results

### 3.1. Effect of Thermal Manipulation (TM) on the Hatchability Rate and Post-Hatch Body Weights of Broiler Chickens

It was found that TM during broiler embryogenesis led to a significantly lower hatchability rate (hatchability of control group = 93.23%; hatchability of TM group = 87.31%; *p*-value < 0.05). Furthermore, TM did not significantly change the body weight of broiler chickens, except on post-hatch day 35, when the body weights of the TM group were significantly higher than those of control group (*p*-value < 0.05) (Figure 1).

### 3.2. Effect of Thermal Manipulation (TM) and Acute Heat Stress (AHS) on mRNA Levels of Splenic Cytokines

The impact of AHS (40 °C for 1, 3, 5, and 7 h on post-hatch day 28) on the splenic mRNA expressions of IFN-α, IFN-β, IFN-γ, IL-4, IL-8, IL-15, IL-16, IL-17, and IL-18 in TM and control broilers are shown in Figure 2, Figure 3 and Figure 4.

#### 3.2.1. IFN-α

Before AHS (at 0 h), basal mRNA expression of IFN-α did not significantly differ between the TM and control groups. However, IFN-α expression was significantly lower in the TM group, compared to the control group, after 1 and 5 h of AHS. In the control group, IFN-α expression significantly increased after 3 and 5 h of AHS compared to the basal level, while, in TM, expression significantly increased after 1 and 3 h of AHS.

#### 3.2.2. IFN-β

Before AHS (at 0 h), basal mRNA expression of IFN-β was significantly lower in the TM group, compared to the control group. During AHS, IFN-β expression was significantly higher after 3 h and significantly lower after 5 h in TM chicks, compared to controls. Relative to the basal level, IFN-β expression significantly decreased after 1 and 3 h of AHS in the control group, while, in the TM group, the expression did not significantly change.

#### 3.2.3. IFN-γ

Before AHS (at 0 h), basal mRNA expression of IFN-γ did not significantly differ between the TM and control groups. During AHS, IFN-γ expression was significantly higher in TM chicks, compared to controls, after 1 h. Compared to the basal level, IFN-γ expression did not significantly change in the control group during AHS, while, in the TM group, expression significantly increased after 1, 3, and 7 h of AHS.

#### 3.2.4. IL-4

Before AHS (at 0 h), basal IL-4 mRNA expression did not significantly differ between the TM and control groups. During AHS, IL-4 expression was significantly lower after 1 h and significantly higher after 5 h in TM chicks, compared to controls. Compared to the basal level, IL-4 expression significantly decreased in controls after 5 h of AHS, while, in the TM group, expression significantly decreased after 1 h of AHS.

#### 3.2.5. IL-8

Before AHS (at 0 h), basal mRNA expression of IL-8 did not significantly differ between the TM and control groups. However, IL-8 expression was significantly higher in TM chicks, compared to controls, after 3 and 5 h and significantly lower after 7 h of AHS. Compared to the basal level, IL-8 expression in controls significantly increased after 5 and 7 h of AHS, while it significantly increased in TM chicks after 1, 3, and 5 h of AHS.

#### 3.2.6. IL-15

Before AHS (at 0 h), basal mRNA expression of IL-15 was not significantly different between the TM and control groups. During AHS, IL-15 expression was significantly higher in TM chicks, compared to controls, after 1 and 3 h. Compared to the basal level, IL-15 expression in controls significantly increased after 3 and 5 h of AHS, while expression significantly increased after 1, 3, and 5 h of AHS in TM chicks.

#### 3.2.7. IL-16

Before AHS (at 0 h), basal mRNA expression of IL-16 did not significantly differ between the TM and control groups. During AHS, IL-16 expression was significantly higher in TM chicks, compared to controls, after 3 h. Relative to the basal level, IL-16 expression in controls did not significantly change during AHS, while the expression significantly increased in TM chicks after 1 and 3 h of AHS.

#### 3.2.8. IL-17

Before AHS (at 0 h), basal mRNA expression of IL-17 was not significantly different between the TM and control groups. Furthermore, no significant difference was observed in IL-17 expression between the TM and control groups during AHS. Compared to the basal level, IL-17 expression significantly increased after 1 and 3 h of AHS in both the TM and control groups.

#### 3.2.9. IL-18

Before AHS (at 0 h), basal mRNA expression of IL-18 was not significantly different between the TM and control groups. During AHS, IL-18 expression was significantly higher in TM chicks, compared to controls, after 1 and 3 h. Relative to the basal level, IL-18 expression significantly increased in controls after 3, 5, and 7 h of AHS, while the expression significantly increased after 1 and 3 h of AHS in the TM group.

### 3.3. Effect of Thermal Manipulation (TM) and Acute Heat Stress (AHS) on IL-4, IL-8, and IFN-γ Plasma Levels

The impact of AHS (40 °C for 1, 3, 5, and 7 h on post-hatch day 28) on IL-4, IL-8, and IFN-γ plasma levels in TM and control groups of broiler chickens is shown in Table 3.

#### 3.3.1. IL-4

Before AHS (at 0 h), no significant difference was observed in the plasma level of IL-4 between the TM and control groups. During AHS exposure, IL-4 plasma level was significantly lower after 3 h and significantly higher after 5 h in TM chicks, compared to that in controls. In the control group, IL-4 plasma levels significantly increased after 3 h of AHS compared to the level at 0 h, while the level did not significantly change in the TM group during AHS exposure.

#### 3.3.2. IL-8

Before AHS (at 0 h), the plasma level of IL-8 was significantly lower in TM chicks, compared to that in controls. During AHS exposure, the level was significantly lower after 1 h and significantly higher after 3 and 5 h of AHS in TM chicks, compared to controls. In the control group, IL-8 plasma levels significantly increased after 7 h, whereas the level significantly increased in the TM group after 3, 5, and 7 h of AHS, compared to the level at 0 h.

#### 3.3.3. IFN-γ

Before AHS (at 0 h), the plasma level of IFN-γ was significantly lower in TM chicks, compared to that in controls. IFN-γ plasma levels were also significantly lower in TM chicks than in controls after 1 and 7 h of AHS, but no significant difference was observed after 3 and 5 h. In the control group, IFN-γ plasma levels did not significantly change during AHS exposure, but they significantly increased in the TM group after 3 and 5 h of AHS, compared to the level at 0 h of AHS.

## 4. Discussion

The aim of the current study was to examine the impact of TM (39 °C and 65% RH for 18 h daily during ED 10 to 18) and subsequent AHS on the splenic mRNA expression of the IFN-α, IFN-β, IFN-γ, IL-4, IL-8, IL-15, IL-16, IL-17, and IL-18 cytokines, as well as on the plasma levels of IFN-γ, IL-4, and IL-8, in broiler chickens. As previously mentioned, cytokines are signaling molecules which play major roles in immune defense and development, as well as tissue regeneration and repair (including wound-healing processes) [28,29].

In the current study, the hatchability rate was significantly lower in the TM group. Although some studies reported no significant impact on hatchability rate [30,31,32], others have shown that TM affects (either by decreasing or increasing) the hatchability rate in broiler chickens [16,22,33,34]. Such contradictory findings could be attributed to the breed, age of the chicks and their mothers, or the humidity and temperature of incubation.

The current findings revealed that TM did not affect the weights of chicks during post-hatch life, except on post hatch day 35, when TM significantly increased body weight. Similarly, previous studies reported that chicks incubated at 39.5 °C for 3 h/d during ED 8–10 had unaffected weights during post-hatch life. However, other studies reported that increasing the incubation temperature to 38.5 °C during ED 16–18 and to 39 °C 18 h/d during ED 12–18 resulted in increased broiler body weights [34,35]. Furthermore, egg incubation at 39.5 °C for 12 h/d during ED 7–16 led to increased myofiber diameters and enhanced muscle growth, up to 35 days of age [36]. In addition, increased incubation temperature has been reported to improve pectoral muscle weights and myoblast proliferation during the post-hatch lives of chickens [34,37,38,39].

In this study, TM did not affect basal IFN-α mRNA levels, but significantly reduced the basal mRNA level of IFN-β. Furthermore, IFN-α expression in both the TM and control groups was significantly increased during AHS, but the expression in the TM group was lower than that of control chickens after 1 and 5 h. IFN-β expression significantly decreased after 1 and 3 h of AHS in the control group, while, in the TM group, IFN-β expression did not change. IFN-α and IFN-β are type-1 interferons that have a main function in the defense against viral infections [7,28]. No previously published data could be found regarding the expression of type-1 interferons under heat stress conditions in chickens. However, in mammals, heat stroke has been reported to increase IFN-α levels [40]. The altered IFN-α expression dynamics and the unchanged expression of IFN-β in TM chicks could be seen as a mitigation measure, in response to AHS.

The current results showed that the expression of IFN-γ significantly increased in the TM group, but was not significantly changed in the control group, during AHS. Furthermore, IFN-γ plasma levels were significantly lower in the TM group at 0 h, compared to the controls, and AHS increased plasma IFN-γ levels only in TM chicks. Correspondingly, previous studies have reported increased IFN-γ expression in chickens during heat stress conditions [14,41]. In contrast, another study has showed that heat-stressed broiler chickens possessed lower levels of IFN-γ expression [5]. IFN-γ has an important role in macrophage activation, along with some antiviral activity, and is mainly produced by Th1 cells [7,28]. In ducklings, TM led to a significant increase in IFN-γ expression during a post-hatch LPS challenge, suggesting that pre-hatch exposure to thermal stress had a beneficial effect on the immune response to inflammatory stimulus during post-hatch life [27]. On the other hand, the present results showed that IL-4 expression significantly decreased after 1 h of AHS in TM chicks and after 5 h in control chicks. In addition, 3 h of AHS increased IL-4 plasma levels in control chicks only. IL-4 is a Th2-produced cytokine which has a major function in antibody-mediated immunity [7]. It has been found that heat stress led to increased IL-4 expression in broiler chickens [5]. Decreased IL-4 expression might be due to increased IFN-γ expression, since the immune system is functionally polarized, into either the pathways associated with a Th1 response (IFN-γ) or into the pathways associated with a Th2 response (IL-4) [42,43].

In this study, AHS significantly increased IL-15 expression in both the TM and control groups; however, IL-15 expression dynamics were only improved in the TM group. IL-15 plays a role in the growth and proliferation of T-cells, B-cells, intestinal epithelium, and natural killer cells, as well as stimulating other cells to produce cytokines [28]. This is the first study that has evaluated the effects of TM and AHS on IL-15 expression. Considering the aforementioned functions of IL-15, our results might suggest that the chicks’ immune system respond to AHS in order to reach homeostasis.

The present results showed that the expressions of IL-8, IL-17, and IL-18 were significantly increased during AHS in both the TM and control chicks, but the expression dynamics of IL-8 and IL-18 were more rapid in TM chicks. The plasma levels of IL-8 were significantly lower in TM chicks, compared to that in controls, at 0 h and were significantly increased after 3 and 5 h of AHS in TM chicks. However, in control chicks, IL-8 plasma levels only significantly increased after 7 h of AHS. Additionally, IL-16 expression significantly increased in TM chicks during AHS but did not change in control chicks. IL-8, IL-16, IL-17, and IL-18 are pro-inflammatory cytokines which have critical functions in innate immunity and the induction of the acute phase response [7,28]. Pro-inflammatory cytokines, especially IL-8, play an important role in the repair and regeneration of injured tissues [28,29,44,45,46,47,48]. Previous studies have shown that heat stress increases IL-8 expression in both chickens and mammals, but decreased IL-8 plasma levels in chickens [12,13,40]. Moreover, it has been reported that heat stress did not affect the expression of IL-18 in broiler chickens [5], but modulated IL-18 levels in mammals [12]. Furthermore, stress induced by corticosterone administration led to increased IL-8 and IL-18 expression in chickens [49,50]. No data could be found in the literature on the expressions of IL-16 and IL-17 during heat stress conditions. However, the present results showed improved pro-inflammatory cytokine expression during AHS in TM chickens.

## 5. Conclusions

The results of this study may suggest that TM at 39 °C and 65% RH for 18 h daily during ED 10 to 18 has long-term effects on broiler immunity, the latter of which is characterized by modulating splenic mRNA expression dynamics of cytokines during post-hatch AHS exposure. Furthermore, TM may improve heat tolerance acquisition by increasing the expression of signaling proteins important to tissue stability and in repair mechanisms that are employed during and/or after heat stress recovery.

## Figures and Tables

**Figure 1 animals-09-00499-f001:**
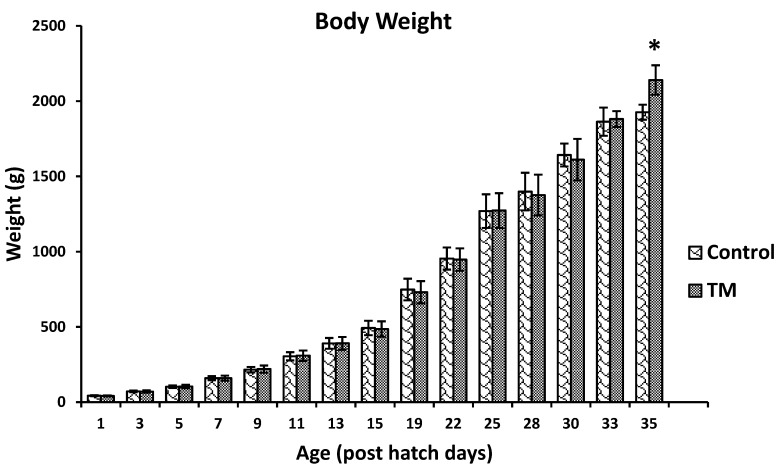
Effect of thermal manipulation (TM) during broiler embryogenesis on post-hatch body weights. ***** Within the same day, values ± SD of TM and control groups are significantly different (*p* < 0.05).

**Figure 2 animals-09-00499-f002:**
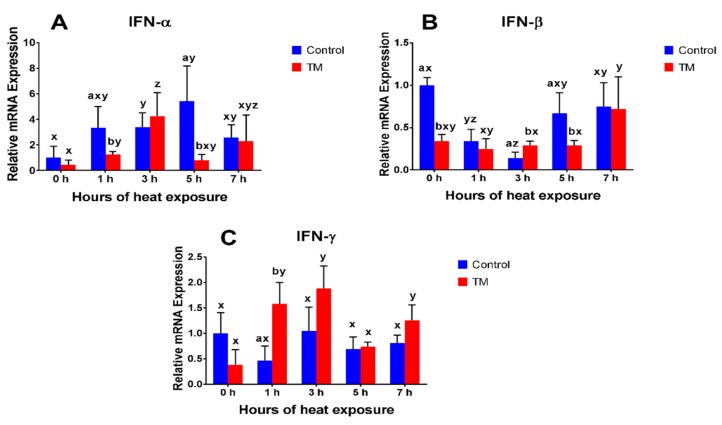
Impact of AHSF at 40 °C for 1, 3, 5, and 7 h on post-hatch day 28 on splenic mRNA levels of (**A**) IFN-α, (**B**) IFN-β, and (**C**) IFN-γ in broiler chickens subjected to TM (*n* = 5). The values in the chart indicate folds changes of mRNA levels compared to the control groups at 0 h before AHS. ^a,b^ Within the same point of time, values ± SD of TM and control groups with different superscripts are significantly different (*p* < 0.05). ^x–y^ Within the same treatment group and between different points of time, values ± SD with different superscripts are significantly different (*p* < 0.05).

**Figure 3 animals-09-00499-f003:**
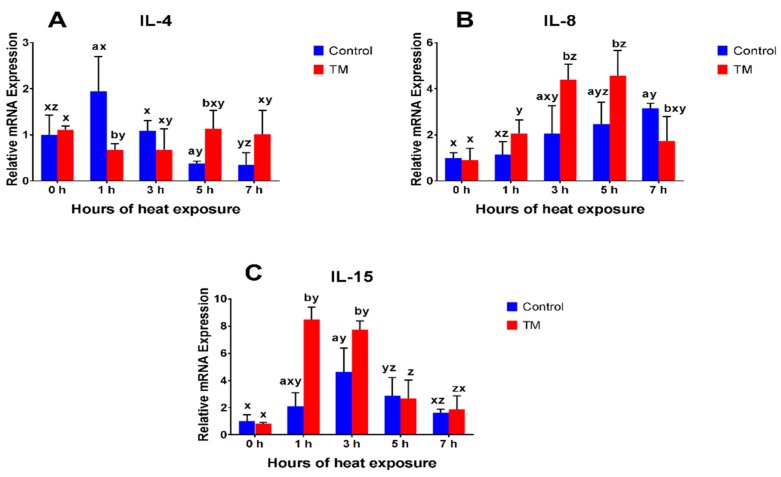
Impact of AHS at 40 °C for for 1, 3, 5, and 7 h on post-hatch day 28 on the splenic mRNA levels of (**A**) IL-4, (**B**) IL-8, and (**C**) IL-15 in broiler chickens subjected to TM (*n* = 5). The values in the chart indicate folds of mRNA levels compared to the control groups at 0 h before AHS. ^a,b^ Within the same point of time, values ± SD of TM and control groups with different superscripts are significantly different (*p* < 0.05). ^x–z^ Within the same treatment group and between different points of time, values ± SD with different superscripts are significantly different (*p* < 0.05).

**Figure 4 animals-09-00499-f004:**
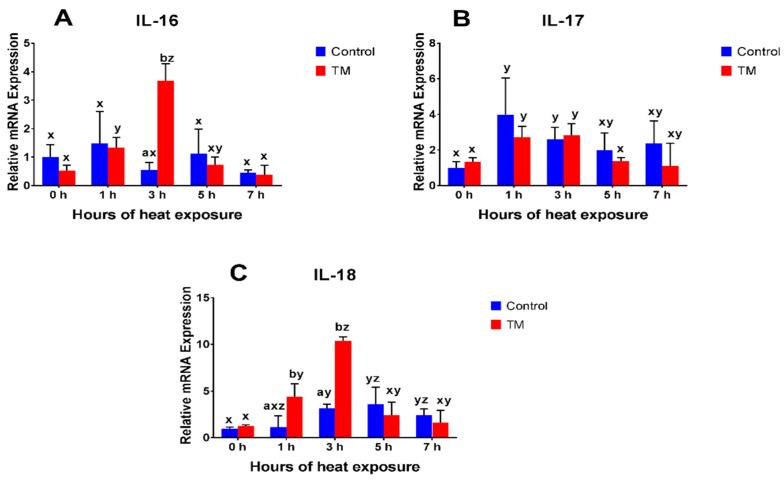
Effects of AHS at 40 °C for 1, 3, 5, and 7 h on post-hatch day 28 on the splenic mRNA levels of (**A**) IL-16, (**B**) IL-17, and (**C**) IL-18 in broiler chickens subjected to TM (*n* = 5). The values in the chart indicate folds of mRNA levels compared to the control groups at 0 h before AHS. ^a,b^ Within the same point of time, values ± SD of TM and control groups with different superscripts are significantly different (*p* < 0.05). ^x–z^ Within the same treatment group and between different points of time, values ± SD with different superscripts are significantly different (*p* < 0.05).

**Table 1 animals-09-00499-t001:** Composition of the experimental diets.

Ingredient (% As fed)	Starter Diet	Grower Diet
	Day 1–21	Day 22–35
Corn	57	64
Soybean meal	32	25
Concentrate	6	5.5
Soybean oil	1	2
Dicalcium phosphate	1.1	0.90
Limestone	1.5	1.5
Mineral-vitamin premix ^a^	0.75	0.50
Salt (NaCl)	0.25	0.25
DL-Methionine	0.23	0.21
L-Lysine HCl	0.17	0.14
**Calculated nutritive value (g kg^−1^) DM ^b^**		
AME_n_ (kcal kg^−1^) ^c^	3120	3185
Crude protein (DM %)	22.3	19.2
Ether Extract (DM %)	4.9	7.8

^a^ Provided the following per kilogram of diet: vitamin A, 11,166 IU; cholecalciferol, 2500 IU; vitamin E, 80 mg; menadione, 2.50 mg; vitamin B_12_, 0.02 mg; folic acid, 1.17 mg; choline, 379 mg; D-pantothenic acid, 12.50 mg; riboflavin, 7.0 mg; niacin, 41.67 mg; thiamine, 2.17 mg; D-bioyin, 0.18 mg; pyridoxine, 4.0 mg; ethoxyquin, 0.09 mg; Mn (MnO_2_), 73 mg; Zn (ZnO), 55 mg; Fe (FeSO_4_), 45 mg; Cu (CuSO_4_), 20 mg; I (CaI_2_O_6_), 0.62 mg; and Se (Na_2_SeO_3_), 0.3 mg. ^b^ DM stands for dry matter content. ^c^ AME_n_ stands for Apparent metabolizable energy.

**Table 2 animals-09-00499-t002:** Primer sequences used in the real-time RT-qPCR analysis.

Gene	Sequence (5′-3′)
**28S rRNA**	F: CCTGAATCCCGAGGTTAACTATT
	R: GAGGTGCGGCTTATCATCTATC
**IL-4**	F: GAGAGCATCCGGATAGTGAATG
	R: TGTGGAGGCTTTGCATAAGAG
**IL-8**	F: CTGCGGTGCCAGTGCATTA
	R: AGCACACCTCTCTTCCATCC
**IL-15**	F: GTGGTCAGACGTTCTGAAAGAT
	R: CAGGTTCCTGGCATTCTATATCC
**IL-16**	F: GGAACAAAGCAGCCCAGTTC
	R: GGCTGTGGTGTGCACCTGTA
**IL-17**	F: CTCCGATCCCTTATTCTCCTC
	R: AAGCGGTTGTGGTCCTCAT
**IL-18**	F: AGGTGAAATCTGGCAGTGGAAT
	R: TGAAGGCGCGGTGGTTT
**IFN-α**	F: ATGCCACCTTCTCTCACGAC
	R: AGGCGCTGTAATCGTTGTCT
**IFN-β**	F: CCTCAACCAGATCCAGCATTA
	R: TAGTTGTTGTGCCGTAGGAAG
**IFN-γ**	F: CAAGTCAAAGCCGCACATC
	R: CGCTGGATTCTCAAGTCGTT

**Table 3 animals-09-00499-t003:** Effect of thermal manipulation (TM) at 39 °C and 65% RH for 18 h daily during embryonic days 10–18 on IL-4, IL-8, and IFN-γ plasma levels in broiler chickens during acute heat stress (AHS) exposure.

IL-4 (pg/mL)	0 h	1 h	3 h	5 h	7 h
**Control**	227.01 ± 2.81 ^ax^	301.19 ± 74.13 ^axy^	304.67 ± 8.61 ^by^	246.70 ± 17.83 ^bx^	250.30 ± 44.22 ^axy^
**TM**	205.21 ± 84.48 ^ax^	219.88 ± 23.90 ^ax^	243.25 ± 33.06 ^ax^	284.74 ± 23.06 ^ax^	248.69 ± 72.82 ^ax^
**IL-8 (pg/mL)**	**0 h**	**1 h**	**3 h**	**5 h**	**7 h**
**Control**	30.35 ± 4.68 ^bx^	33.48 ± 5.26 ^bxy^	33.94 ± 8.64 ^bxy^	30.91 ± 6.20 ^bxy^	38.17 ± 7.36 ^ay^
**TM**	21.37 ± 4.65 ^ax^	26.18 ± 6.91 ^axz^	41.01 ± 3.55 ^ay^	38.72 ± 4.53 ^ay^	31.12 ± 9.04 ^az^
**IFN-γ (pg/mL)**	**0 h**	**1 h**	**3 h**	**5 h**	**7 h**
**Control**	179.18 ± 48.18 ^bx^	155.88 ± 63.06 ^bx^	183.36 ± 84.59 ^ax^	179.39 ± 69.86 ^ax^	204.39 ± 41.40 ^bx^
**TM**	99.58 ± 46.31 ^ax^	93.23 ± 45.77 ^ax^	189.23 ± 32.46 ^ay^	199.92 ± 58.76 ^ay^	108.00 ± 52.19 ^ax^

^a,b^ Within the same column, values with different superscripts differ significantly (*p*-value < 0.05). ^x–z^ Within the same row, values with different superscripts differ significantly (*p*-value < 0.05).

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
