# Peer review of "Effect of Acute Heat Stress on the mRNA Levels of Cytokines in Broiler Chickens Subjected to Embryonic Thermal Manipulation"

_animals, 2019, doi:10.3390/ani9080499_

Round 1
Reviewer 1 Report
This study is about embryonic heat stress followed by post hatch heat stress on panel of cytokines expression.My biggest concern in this study is about the experimental design. The authors gave the total number of eggs and didn’t mention how did they stratified the embryos. Number of birds per treatment/replication is missing. It is not clear about their experimental design. Prior, the birds were treated with AHS, birds were received 2 vaccines at 3 different time points d8, d15 and d20. This vaccination also cause additional stress to the birds. So the control groups are not technically “real control” birds. This vaccination might be confounding factors especially secondary heat stress on cytokines expression. Authors should justify the cytokines effect they see in this study came only from TM treatment not from the vaccination/vaccination stress. Vaccination induce stress as well as cytokine expression. In addition, if they have presented the production parameters data (growth performance) it could be easy to understand whether embryonic thermal manipulation helped post-hatch heat stress. In the discussion part, please explain whether embryonic TM induce inflammation or immunosuppression based on the cytokines you measured.
Line 103: please explain why you used different relative humidity between 2 incubators and how did you set up the eggs? For example number of eggs/tray and location within the incubator.
Line 111: why JUST in capital letters?
Line 113-114: little bit confusion in the sentence. Were they maintained at 21 or 24 degree celcius?
Line 131: please explain whether you used random primers or oligo dt primers you used for making cDNA since 28s primers lack of polyA tail. Can you justify how 28s primers are right housekeeping genes for normalization?. Please justify Embryonic heat stress did not induce epigenetic modification on 28S gene expression
Line 133: please explain hatchability percentage, Number of birds used per treatment, biological replication( n=?) and technical replicate for analysis.
Results section biological replicates per treatment is missing.
It is not clear about your experimental design. Do you have control groups for TM groups. If my understanding is right it is supposed to be 4 treatment groups like this below. However in the graphs 2 groups are missing as well as number of birds used per treatment. Was it one bird/pen or 2 birds per pen. How many replicates( n). Please explain why did you choose 2 way ANOVA since you have 2 treatments per time point. I assume the following is the right experimental model. Control birds from controlled temp and control birds TM treatment are missing.
Con TM
Con AH TM CON TM-AHS
In this study pro inflammatory cytokines as well as IL4 significantly increased. Did this increased expression affect the growth performance/ mortality?.
Author Response
Line 103: please explain why you used different relative humidity between 2 incubators and how did you set up the eggs? For example number of eggs/tray and location within the incubator.
Line 107: Response: the following line was added to explain the different relative humidity (The RH was increased to 65% to avoid an increase in water evaporation from the eggs during TM.
Line 111: why JUST in capital letters?
Line 94: Response: JUST is abbreviation for Jordan University of Science and Technology.
Line 113-114: little bit confusion in the sentence. Were they maintained at 21 or 24 degree celcius?
Line 115-118: Response: sentence was reversed to Room temperature was maintained at 33±1°C and 50-60 % RH during the first week post-hatch and was gradually decreased to 24°C and 50-60 % RH by the end of the third week post-hatch. Starting from post-hatch day 24 until day 28, the room temperature was maintained at 21°C and 50-60 % RH
Line 131: please explain whether you used random primers or oligo dt primers you used for making cDNA since 28s primers lack of polyA tail. Can you justify how 28s primers are right housekeeping genes for normalization?. Please justify Embryonic heat stress did not induce epigenetic modification on 28S gene expression.
Response: we used the random primers.
Response: in my previous manuscript that has been published I got the same comment concerning the House keep gene in my manuscript (Al-Zghoul et al., 2019. Effects of pre-hatch thermal manipulation and post-hatch acute heat stress on the mRNA expression of interleukin-6 and genes involved in its induction pathways in 2 broiler chicken breeds, Poultry Science, 98(4): 1805–1819, https://doi.org/10.3382/ps/pey499). We ordered several reference genes (GAPDH, Beta-actin and G6DH) and repeated the housekeeping gene RT-PCR analysis. We calculated the relative genes expression against the S28 and other reference genes (GAPDH, Beta-actin and G6DH), comparable results were obtained.
Line 133: please explain hatchability percentage, Number of birds used per treatment, biological replication( n=?) and technical replicate for analysis.
Results section biological replicates per treatment is missing.
It is not clear about your experimental design. Do you have control groups for TM groups. If my understanding is right it is supposed to be 4 treatment groups like this below. However in the graphs 2 groups are missing as well as number of birds used per treatment. Was it one bird/pen or 2 birds per pen. How many replicates( n). Please explain why did you choose 2 way ANOVA since you have 2 treatments per time point. I assume the following is the right experimental model. Control birds from controlled temp and control birds TM treatment are missing.
Response: the numbers is added. Line 1004, Line 105, Line 115, Line 134, line 150.
2 way ANOVA was used here as we had two treatment groups (TM vs. control), (AHS at 0 h of AHS vs. 1, 3, 5 and 7 h after AHS).
Reviewer 2 Report
Effect of acute heat stress on the mRNA levels of cytokines in broiler chickens subjected to embryonic thermal manipulation
General comments
The authors proposed to relate thermal manipulation during embryogenesis on cytokine gene transcription in an attempt to understand the mechanisms of heat tolerance in broilers.
The manuscript is well written and addresses an important problem that is likely to become more of a issue with increasing global temperatures.
Specific comments
There is a concern over the lack of information concerning the quality of the extracted RNA and the use of a single reference for housekeeping gene.
The quality of the total RNA needs to be described and a RIN provided to ensure that the amplifications reported are derived from intact mRNA. Please respond with reference to the recommendations of Vermeulen et al., 2011. Measurable impact of RNA quality on gene expression results from quantitative PCR. Nucleic Acids Research, Vol. 39, No. 9 e63. doi:10.1093/nar/gkr065
How was the reference gene selected? Stability determined? How many? Were the amplicons determined to be single sequences for each primer pair/template/sample?
The use of more than one reference or housekeeping gene has been suggested by recent studies (Jacob et al., 2013). The authors state “For establishing a set of reference genes for gene normalization we recommend the use of ideally three reference genes selected by at least three stability algorithms.”
The MIQE guidelines (Bustin et al., 2009) referred to by Jacob et al., 2013 state that “Normalization against a single reference gene is not acceptable unless the investigators present clear evidence for the reviewers that confirms its invariant expression under the experimental conditions described. The optimal number and choice of reference genes must be experimentally determined and the method reported.”
Bustin 2009 MIQE Minimum Information for Publication of Quantitative Real-Time PCR Clinical Chemistry 55:4 611–622
Vermeulen 2011 Measurable impact of RNA quality on gene expression results from quantitative PCR Nucleic Acids Research, 2011, Vol. 39, No. 9 e63
Nolan 2006 Quantification of mRNA using real-time RT-PCR. Nature Protocols Nov. 1(3):1559-1582
Jacob 2013 Careful Selection of Reference Genes Is Required for Reliable Performance of RT-qPCR in Human Normal and Cancer Cell Lines PLoS ONE 8(3): e59180. doi:10.1371/journal.pone.0059180
Author Response
Response: samples were collected and directly snap-freeze in liguid nitrogen in order to prevent any damage for RNA. RNA was isolated from the sample tissues by Direct-Zol kit.
RNA concentrations and purities were first determined using the Biotek PowerWave XS2 Spectrophotometer and followed by Qubit4 to determine the integrity of the RNA. Samples with 260/280 ratio were excluded and RNA isolation was repeated to ensure that staring point of RNA concentration and purity is acceptable.
Before we started the Rt-qPCR analyses, we performed the cDNA synthesis for samples (in duplicate). We performed the real-time RT-PCR in several house keep genes (Beta-actin, S28, GAPDH and G6DH), comparable CT values were obtained. Melting curve analysis confirmed that S28 and beta-actin and GAPDH had similar results. Based on theses results we used the S28 as our house keeping genes. Furthermore, the results from different genes analyses were tested against these house keeping genes, again comparable results were obtained. Similarly, in my previous manuscript that has been published I got the same comment concerning the House keep gene in my manuscript (Al-Zghoul et al., 2019. Effects of pre-hatch thermal manipulation and post-hatch acute heat stress on the mRNA expression of interleukin-6 and genes involved in its induction pathways in 2 broiler chicken breeds, Poultry Science, 98(4): 1805–1819, https://doi.org/10.3382/ps/pey499). We ordered several reference genes (GAPDH, Beta-actin and G6DH) and repeated the housekeeping gene RT-PCR analysis. We calculated the relative genes expression against the S28 and other reference genes (GAPDH, Beta-actin and G6DH), comparable results were obtained.
The house keeping genes used in these experiments were added to the table 2, page 4.
The primer sequences of Beta-actin and GAPDH were added to the table 2.
Beta-Actin | F-TTGTTGACAATGGCTCCGGT R-TCTGGGCTTCATCACCAACG |
GAPDH | ACTGTCAAGGCTGAGAACGG CATTTGATGTTGCTGGGGTC |
Reviewer 3 Report
1. Line 29, please use “Heat stress significantly impacts chicken immunity and cytokine expression. However, the effects of embryonic thermal manipulation…..”
2. Line 55-56, delete “The broiler chicken, Gallus gallus domesticus, is one variety of poultry that is specifically bred and reared for commercial meat production.”
3. Line 56, please use “Exposing the broilers to high environmental temperatures….”
4. In the Materials and Methods section, what’s the age of the breeder flock? What’s the initial range of the egg weights? What’s the average egg weight across the treatment group? With the same initial egg weight?
5. In L114, what’s the “appropriate feed” mean? What’s the nutrient concentration? At lease provide the energy and crude protein level? Was the feed in mash form?
6. Did the author measure the “mean egg shell temperature” or the “yolk free wet embryo body weight of the embryo” in the study?
7. In line 348-349 “The results of this study suggest that TM at 39oC and 65% RH for 18 h daily during ED 10 to 18 has long-term effects on broiler immunity”: Since the study was a short period trial and did not include the immune parameter results for a full grow-out period, it is bold to draw such conclusion based on the limited data. Please make conclusion based on what has been found in the current study. It could be more specific instead of such a general conclusion sentence.
Author Response
1. Line 29, please use “Heat stress significantly impacts chicken immunity and cytokine expression. However, the effects of embryonic thermal manipulation…..”
Line 31: Response: “Heat stress significantly impacts chicken immunity and cytokine expression. However, the effects of embryonic thermal manipulation…..” is used.
2. Line 55-56, delete “The broiler chicken, Gallus gallus domesticus, is one variety of poultry that is specifically bred and reared for commercial meat production.”
Response: “The broiler chicken, Gallus gallus domesticus, is one variety of poultry that is specifically bred and reared for commercial meat production.” Is deleted.
3. Line 56, please use “Exposing the broilers to high environmental temperatures….”
Line 57: Response: “Exposing the broilers to high environmental temperatures….” Is used.
4. In the Materials and Methods section, what’s the age of the breeder flock? What’s the initial range of the egg weights? What’s the average egg weight across the treatment group? With the same initial egg weight?
Response Line 101: Response: Six hundred (600) fertile eggs belonging to the Hubbard broiler strain were obtained from a certified breeder based in Irbid, Jordan. Eggs were inspected for any breakage or abnormality, and eggs were excluded if they were broken, deformed, round, wrinkled, or outside the normal size range (55g < eggs < 70g).
5. In L114, what’s the “appropriate feed” mean? What’s the nutrient concentration? At lease provide the energy and crude protein level? Was the feed in mash form?
Response: line 122 a table 1 is added to described the feed rations.
6. Did the author measure the “mean egg shell temperature” or the “yolk free wet embryo body weight of the embryo” in the study?
Response: we did that, and it is reported in the following articles where we used the same material and methods (Al-Zghoul et al., 2019, Expression of digestive enzyme and intestinal transporter genes during chronic heat stress in the thermally manipulated broiler chicken. Poultry Science, pez249, https://doi.org/10.3382/ps/pez249), Al-Zghoul et al., 2019. Effects of pre-hatch thermal manipulation and post-hatch acute heat stress on the mRNA expression of interleukin-6 and genes involved in its induction pathways in 2 broiler chicken breeds. Poultry Science, Volume 98, Issue 4, April 2019, Pages 1805–1819, https://doi.org/10.3382/ps/pey499). We did not measure the egg shell temperature.
7. In line 348-349 “The results of this study suggest that TM at 39oC and 65% RH for 18 h daily during ED 10 to 18 has long-term effects on broiler immunity”: Since the study was a short period trial and did not include the immune parameter results for a full grow-out period, it is bold to draw such conclusion based on the limited data. Please make conclusion based on what has been found in the current study. It could be more specific instead of such a general conclusion sentence.
Line 354-359 Response: the conclusion is modified as following (The results of this study may suggest that TM at 39°C and 65% RH for 18 h daily during ED 10 to 18 has long-term effects on broiler immunity, the latter of which is characterized by modulating splenic mRNA expression dynamics of cytokines during post-hatch AHS exposure. Furthermore, TM may improve heat tolerance acquisition by increasing the expression of signaling proteins important to the tissue stability and in repair mechanisms that are employed during and/or after heat stress recovery).
Round 2
Reviewer 2 Report
All areas addressed.